# Interrogating Genomes and Geography to Unravel Multiyear Vesicular Stomatitis Epizootics

**DOI:** 10.3390/v16071118

**Published:** 2024-07-11

**Authors:** John M. Humphreys, Phillip T. Shults, Lauro Velazquez-Salinas, Miranda R. Bertram, Angela M. Pelzel-McCluskey, Steven J. Pauszek, Debra P. C. Peters, Luis L. Rodriguez

**Affiliations:** 1Foreign Animal Disease Research Unit, Agricultural Research Service, U.S. Department of Agriculture, Plum Island Animal Disease Center (PIADC) and National Bio Agro Defense Facility (NBAF), Manhattan Kansas, KS 66502, USA; lauro.velazquez@usda.gov (L.V.-S.); miranda.bertram@usda.gov (M.R.B.); luis.rodriguez@usda.gov (L.L.R.); 2Arthropod-Borne Animal Disease Research Unit, Agricultural Research Service, U.S. Department of Agriculture, Manhattan, KS 66502, USA; phillip.shults@usda.gov; 3Veterinary Services, Animal and Plant Health Inspection Service (APHIS), U.S. Department of Agriculture, Fort Collins, CO 80526, USA; angela.m.pelzel-mccluskey@usda.gov; 4Foreign Animal Disease Diagnostic Laboratory, National Veterinary Services Laboratories, Animal and Plant Health Inspection Service (APHIS), Plum Island Animal Disease Center (PIADC), U.S. Department of Agriculture, Orient, NY 11957, USA; steve.pauszek@usda.gov; 5Office of National Programs, Agricultural Research Service, U.S. Department of Agriculture, Beltsville, MD 20705, USA; deb.peters@usda.gov

**Keywords:** vesicular stomatitis, vector-borne diseases, livestock, phylogenetics, spatial epidemiology

## Abstract

We conducted an integrative analysis to elucidate the spatial epidemiological patterns of the Vesicular Stomatitis New Jersey virus (VSNJV) during the 2014–15 epizootic cycle in the United States (US). Using georeferenced VSNJV genomics data, confirmed vesicular stomatitis (VS) disease cases from surveillance, and a suite of environmental factors, our study assessed environmental and phylogenetic similarity to compare VS cases reported in 2014 and 2015. Despite uncertainties from incomplete virus sampling and cross-scale spatial processes, patterns suggested multiple independent re-invasion events concurrent with potential viral overwintering between sequential seasons. Our findings pointed to a geographically defined southern virus pool at the US–Mexico interface as the source of VSNJV invasions and overwintering sites. Phylodynamic analysis demonstrated an increase in virus diversity before a rise in case numbers and a pronounced reduction in virus diversity during the winter season, indicative of a genetic bottleneck and a significant narrowing of virus variation between the summer outbreak seasons. Environment–vector interactions underscored the central role of meta-population dynamics in driving disease spread. These insights emphasize the necessity for location- and time-specific management practices, including rapid response, movement restrictions, vector control, and other targeted interventions.

## 1. Introduction

Vesicular stomatitis (VS) is a vector-borne livestock disease caused by the vesicular stomatitis virus (VSV), an RNA virus of the Rhabdovirus family. The most common serotype responsible for VS outbreaks in the United States (US) is the Vesicular Stomatitis New Jersey virus (VSNJV) [1]. The disease is endemic to southern Mexico, where clinical cases associated with several viral lineages are reported annually. However, some VSNJV lineages spread to the southern US, producing epizootic outbreaks at approximately decadal intervals [2,3,4]. Due to reduced productivity and movement restrictions, VS is economically significant to the cattle, swine, and equine industries [5,6,7,8]. Moreover, VS induces considerable concern, as its clinical presentation in cattle and swine is indistinguishable from that of foot-and-mouth disease (FMD), which is an economically devastating livestock infection that was successfully eradicated from the US [9].

Historically, VS outbreaks in the United States have followed a decadal pattern, with disease occurrences spanning two to three consecutive years, interspersed with 8–10 years of quiescence [10,11]. Considerable debate surrounds the mechanisms leading to VSNJV emergence in the US and the factors that contribute to persistence or possible re-invasion during sequential outbreak years. The cyclic pattern shown during US outbreaks has been succinctly characterized by an “incursion–expansion” hypothesis, which posits epidemics as being initiated by virus introductions from Mexico followed by overwintering, geographic expansion, and eventual local extinction in the US within one to three years [3,10,12,13].

Support for the incursion–expansion hypothesis has been bolstered by prior phylogeographic analyses that traced US-collected VSNJV back to endemic viral lineages from Mexico [14,15,16]. These incursions were observed to be followed by short-term persistence or overwintering before showing increased geographic spread during the successive spring and summer seasons. Research has found that US epidemic strains hold close phylogenetic relationships with Mexican antecedents, establishing a latitudinal genetic gradient, with southernmost US isolates displaying higher homology to Mexican strains than those samples collected further north [16]. Evidence for VSNJV overwintering has likewise been shown. Following a 2004–2006 outbreak, for example, VSNJV phylogenetic [15] and spatial analyses [7] demonstrated that US viral isolates grouped or clustered with samples from the previous year, indicating the ability of the virus to locally persist within US host and vector populations. These findings demonstrate that virus evolution and genetic diversity are key to understanding VSNJV transmission across diverse ecological landscapes and host–vector populations.

In addition to genetic considerations, environmental factors play a pivotal role in VSNJV epidemiology. Studies from South, Central, and North America have demonstrated the influence of regional ecology and climatic variation on VSNJV genotype distribution [2,17,18]. Given the tight nexus between environmental conditions and poikilothermic arthropod abundance, physiologic tolerances, and feeding behavior, insect vectors have been identified as the most probable intermediaries for moving viruses between livestock hosts and dispersed host populations [19,20,21]. Environmental factors such as precipitation, temperature, vegetation presence, and proximity to water bodies have all been proposed as contributors to vector-driven transmission in the US [11,13,22]. Despite gaps in our understanding of precise vector–host associations, the isolation of VSNJV from diverse arthropods implies that the vector community contributes to disease propagation across a myriad of species-specific routes [4,18,23,24].

VSNJV transmission is shaped by a mixture of virus genetic adaptation, geographically heterogeneous vector–host populations, and the environmental variables that influence vector abundance and behavior. Further complexity arises from the meta-population dynamics that connect these factors across time and space [25,26,27]. Spatial epidemiology offers a unifying framework for interpreting disease spread patterns across different geographies, as it allows researchers to parse out the contributions of vector, host, and environmental interactions in determining transmission risk [27,28,29,30]. Spatial epidemiology is essential for understanding the episodic nature of VS outbreaks and disentangling the mechanisms that drive disease absence, emergence, and re-emergence in the US [14].

Velazquez-Salinas et al. [16] described the emergence of epizootic VSNJV lineage 1.1 in the US; this lineage was documented as circulating in central and northern Mexico between 2006 and 2009 before reaching the southern US in 2012. A preliminary phylogenetic analysis by Velazquez-Salinas et al. [31] confirmed that this lineage remained active in the US until at least 2015. Notably, the 2014–2015 VS outbreak in the US featured a significant increase in cases, highlighting the need to understand VSNJV transmission dynamics. To elucidate the 2014–15 VS outbreak cycle, we integrated georeferenced VSNJV genomics data, confirmed VS case records, and a suite of environmental variables. Our study aimed to discern whether VS cases reported across 2014 and 2015 indicated viral overwintering or repeated virus introductions. Our adoption of an innovative analytical pipeline, which melded phylogeographic and phylodynamic analyses with spatiotemporal modeling, permitted us to navigate the challenges of disease detection and sample collection biases to enhance understanding of both the broad-scale and localized spatial processes associated with outbreaks. Our findings highlight the necessity for refined surveillance protocols attuned to nuanced VSNJV transmission dynamics, with insights from our study informing targeted control and intervention strategies to mitigate VS outbreaks. Our research contributes to a more robust framework for managing and understanding vector-borne diseases by elucidating VS transmission’s spatial and temporal features.

## 2. Materials and Methods

### 2.1. Study Area

The project area included more than 4.25 million km^2^ in the western US and northern Mexico (Figure 1). Portions of the study area in the US included the states of Texas (TX), New Mexico (NM), Arizona (AZ), Oklahoma (OK), Kansas (KS), Colorado (CO), Utah (UT), Nebraska (NE), Wyoming (WY), Idaho (ID), Montana (MT), South Dakota (SD), and North Dakota (ND). In Mexico, the Sonora and Chihuahua states were included, as were the majority of Coahuila and minor portions of Sinaloa, Durango, Nuevo Leon, and Tamaulipas.

### 2.2. Virus and Disease Detection Data

VSNJV viral isolates (*n* = 240) sampled during the US 2014–15 outbreak cycle were combined with a 2013 VSNJV isolate sample from Chihuahua, Mexico (*n* = 1) for genomics analysis (Figure 2). US isolates were sampled from 51 bovine and 189 equine hosts and obtained from the USDA Foreign Animal Disease Diagnostic Laboratory (FADDL) at Plum Island Animal Disease Center and the USDA APHIS National Veterinary Services Laboratories (NVSL). The single Chihuahua isolate (bovine) was obtained from the Exotic Animal Disease Laboratory in Mexico. Samples from FADDL were received as passage 1 supernatant recovered from Vero, porcine kidney (IBRS-2), or lamb kidney (LK) cell lines. Samples from NVSL were received as passage 1 or 2 supernatant recovered from baby hamster kidney (BHK) cell lines. US-sampled isolates included premises-level geographic coordinates indicating sample collection location; however, the Mexico isolate did not include coordinates. Therefore, the center of Chihuahua served as the basis for geographic analysis (28.63° North, 106.07° West).

In addition to genetic samples, the study incorporated veterinarian-confirmed detections of VS animal disease (*n* = 1258) documented during the 2014–15 outbreak (Figure 2). USDA APHIS Veterinary Services provided VS detection data and, like the US isolates, included timestamped, point-level coordinates reflecting the private residence, ranch, farm, or business location (premises-level) where the clinical onset of VS was diagnosed. Premises with confirmed VS cases were reported to maintain equid (1061 premises), bovid (182 premises), and both equid and bovid (15 premises) livestock.

### 2.3. Sequencing and Alignment

RNA was extracted from RT-PCR-positive samples with cycle threshold (Ct) values of up to 26.99. According to the manufacturer’s instructions, DNA was depleted from the extracted samples using an RNase-free DNase kit (Ambion, Austin, TX, USA). First-strand cDNA synthesis was performed using a SuperscriptTM II Reverse Transcriptase kit (Invitrogen, Carlsbad, CA, USA) with random hexamer primers and one conserved VSV intergenic region-specific primer (5). Subsequently, a NEBNext R Ultra Non-Directional RNA Second Strand Synthesis Module was utilized to generate the second strand of cDNA. The cDNA was purified using AmPure XP beads (Beckman Coulter, Brea, CA, USA) and cDNA libraries were created using a Nextera XT DNA Library Preparation kit. Next-generation sequencing (NGS) was conducted on an Illumina NextSeq550 instrument, producing paired-end reads.

These reads were quality trimmed then mapped to two reference sequences (accession no. MG552609 Velazquez-Salinas et al. [31] and accession no. AF473864 Rodriguez et al. [32]) and de novo assembled. Consensus sequences were derived from each of the mappings and the assembly for each sample. These consensus sequences were then aligned, and a final consensus sequence was obtained from the alignment for each sample. All analyses were executed in CLC Workbench v21 using the default parameters.

### 2.4. Phylogeographic and Phylodynamic Inference

To better understand the epidemiological processes that drove the 2014–15 VS outbreaks, phylogenetic analyses were performed to trace lineage dispersal between Mexico and the US in order to reconstruct possible transmission networks among sampled and unsampled hosts and to compare virus diversification rates to a bias-corrected outbreak intensity.

RNA viruses, including VSNJV, evolve rapidly due to their high mutation rates during replication [18]. Their fast-paced evolution makes them prime candidates for studying disease transmission dynamics [33,34]. Phylodynamics combines ecological analysis with phylogenetic inference in a temporal context, enabling examination of the selective pressures, ecological conditions, and demographic changes driving pathogen evolution [35]. This synthesis of genealogical data with variables such as pathogen population size, host distribution, and migration patterns allows for the reconstruction of detailed pathogen transmission pathways.

To trace lineage dispersal, phylogeographic analysis was conducted using a relaxed random walk diffusion model that accommodated different dispersal velocities between lineage branches [36,37,38]. Based on past VSNJV work [10] and model comparison [39], a Hasegawa–Kishino–Yano (HKY) nucleotide substitution model was selected in conjunction with branch evolutionary rates following a relaxed molecular clock as implemented in BEAST v1.10 [40]. Markov chain Monte Carlo (MCMC) chains were run for 250 million generations before applying Tracer software v1.7.2 [41] to verify that convergence, even mixing, and adequate effective sample sizes (ESSs) were achieved. Trees were then summarized as a maximum clade credibility (MCC) tree in Tree Annotator v1.10.4 [40]. Although not used directly, functions described by Dellicour et al. [37] and available in the seraphim r-package [42] served as a guide when extracting spatial and temporal information from the phylogeographic analysis.

As opposed to using the trees produced by phylogeographic analysis to reconstruct transmission networks and to estimate demographic parameters, a birth–death skyline model [43,44] was implemented in BEAST v2.7.1 [45]. In an epidemiological context, births correspond to new infections, whereas deaths represent recovery [46]. Like the phylogeographic analysis, an HKY model was adopted for virus evolution. However, an Ornstein–Uhlenbeck smoothing prior was used, as were Bayesian Integrated Coalescent Epoch PlotS (BICEPS) operators [45] to accelerate the run time and avoid convergence issues. MCMC chains had lengths of 250 million and were evaluated using Tracer. The effective population size (N_*e*_) is correlated to the pathogen genetic diversity, host population size, and total number of infections in epidemiological systems [47,48]; however, N_*e*_ may be biased by uneven sampling [49,50]. To help overcome the issue of preferential sampling, Bayesian nonparametric phylodynamic reconstruction was performed [51,52] to better estimate the VSNJV N_*e*_ trajectory. The estimated N_*e*_ was then exported from the results to be included as a non-linear covariate (predictor variable) during later spatial statistical modeling (see Section 2.5). Trees from the birth–death model were further analyzed to reconstruct transmission networks.

Host transmission networks were inferred from the birth–death model results using TransPhylo v1.4.5 [53]. TransPhylo aided with uncovering probabilities for unsampled VSNJV hosts in the transmission chain and the epidemiological links between sampled and sequenced viruses, thereby providing a more complete picture of the outbreak [53,54]. Transmission model parameter calibration was based on mechanistic compartmental models developed for the US 2014–15 VSNJV outbreak [55]; a gamma prior distribution having a 5.5-day mean and a 2.3-day sd was selected for both the generation and reporting time parameters. The analysis was run for 500,000 iterations with a 25% burn-in on MCMC chains. As with other phylogenetic analyses, convergence, mixing, and ESS were assessed before visualizing with Gephi [56].

### 2.5. Spatial Statistical Model

A Bayesian spatiotemporal model was constructed to quantify VS outbreak intensity by year and location. As used here, the term “intensity” refers to the number of premises-level disease cases estimated per unit area (cases/5 km^2^). We interpret intensity as reflecting the culmination of all epidemiological, ecological, and host demographic processes that contributed to disease emergence, transmission, and spread, including those processes that were unobserved or unmeasured (latent). Intensity can be differentiated from the observed disease occurrences (as illustrated in Figure 2) in that the observed events represent only a sample of the “true” or actual disease prevalence and are assumed to be biased due to imperfect detection, uneven sampling, and incomplete reporting.

To help account for biased detection and reporting, disease intensity was jointly estimated using both VS case occurrences ($Y_{st}$) and sampling density ($\Lambda_{st}$) in a two-part model with shared random effects. Sampling density ($\Lambda_{st}$) is sometimes referred to using the term “intensity”; however, we reserve the term intensity to refer to the jointly estimated case abundance per unit area. More formally, this modeling framework is known as a log-Gaussian Cox process under preferential sampling [57,58,59,60]. Sampling density was estimated in the first component of the two-part model such that
(1)$$\begin{array}{cc} \log(\Lambda_{st}) & =W_{st}\\ \;\;\Lambda_{st} & =\exp\{\beta_{\Lambda }+W_{st}\}\\ W_{st} & \overset{\mathrm{iid}}{∼}N\left(0,Q\left(r,\sigma \right)\right)\\ r & ∼\mathrm{Pr}(250,0.01)\\ \sigma & ∼\mathrm{Pr}(1,0.01) \end{array}$$
The sampling density ($\Lambda_{st}$) at each geographic location *s* (s=1,2,3,…,n) and year *t* (t=2014,2015) was estimated as the exponential of a Gaussian random field ($W_{st}$) plus an intercept ($\beta_{\Lambda }$), where the intercept quantified the study area mean case density and the random field mapped departures from that mean. Because the disease pattern exhibited in 2014 differed from that observed in 2015, year-specific, independent, and identically distributed (iid) random fields were incorporated. The matrices Q used for specifying the random fields ($W_{st}$) were approximated using stochastic partial differential equations [61] with Matérn covariance functions defined based on penalized complexity (PC) priors [62,63]. Random-field PC priors were specified using weakly informative spatial range (*r*) parameters that assumed a 0.01 probability that spatial correlation was approximately zero at 250 km distance and variance parameters (σ) indicated a low probability that variance exceeded a value of 1.0. The distance of 250 km represents about 10% of the study domain at its widest point and was selected after evaluating distances between 25–1000 km in 25 km increments. Although the 250 km threshold was found to be optimal for the study domain on the whole, several isolated localities exhibited elevated autocorrelation due to tightly clustered disease occurrences. To account for disease clusters, a constructed covariate [64] based on case proximity was used to quantify fine-scale spatial structures among disease locations [65] and is discussed further below.

Gaussian random fields ($W_{st}$) from the first model level were then shared with the model’s second level, where they were combined with several additional covariates to better estimate VS case counts at unsampled or unobserved locations. The second part of the joint model was specified as
(2)$$\begin{array}{cc} Y_{st}|\lambda_{st} & ∼\mathrm{Poisson}(\mu_{st})\\ \mu_{st} & =E_{\mathrm{st}}\lambda_{st}\\ log(\mu_{\mathrm{st}}) & =log\left(E_{\mathrm{st}}\right)+log\left(\lambda_{\mathrm{st}}\right)\\ log(\lambda_{\mathrm{st}}) & =\beta_{0}+\sum_{b=1}^{B}\beta_{b}X_{bst}+\gamma_{clust}+\gamma_{river}+\gamma_{ne}+\alpha W_{st}\\ \beta_{b} & =(\beta_{eddi},\beta_{wet},\beta_{temp},\beta_{seas},\beta_{carb},\beta_{shrb},\beta_{elev},\beta_{ndvi})\\ \beta_{b} & ∼N(1,0.001)\\ \gamma_{clust},\gamma_{river} & ∼\mathrm{Pr}(1,0.01)\\ \gamma_{ne} & ∼\mathrm{Pr}(3,0.01) \end{array}$$
where observed VS case occurrences ($Y_{st}$) were conditional on the underlying sampling density ($\lambda_{st}$) and drawn from a Poisson distribution with a mean case number $\mu_{st}$ at location *s* and time *t*. In an epidemiology context, the mean case number ($\mu_{st}$) is the product of the number of *Exposed* (*E*) hosts (hosts ∝ number of premises) and the sampling density ($\lambda_{st}$). By factoring $\mu_{st}$ to the sum of log-Exposure and log-density, $log(E_{\mathrm{st}})$ can be additively modeled as an offset. By comparison, $log(\lambda_{\mathrm{st}})$ was estimated via a linear predictor with several fixed and random effects.

The linear predictor included an intercept ($\beta_{0}$) and coefficient ($\beta_{b}$) design matrix for (linear) fixed effects ($X_{bst}$). Coefficients were estimated for eight fixed effects, including the Evaporative Demand Drought Index ($\beta_{eddi}$) [66,67], a topographic wetness index ($\beta_{wet}$) [68], the mean temperature during the wettest quarter of the year ($\beta_{temp}$), temperature seasonality ($\beta_{seas}$) [69], soil organic carbon (g/kg, $\beta_{carb}$) [70], the proportion of land cover occupied by shrub vegetation ($\beta_{shrb}$) [71], elevation ($\beta_{elev}$) [68], and the mean normalized difference vegetation index between the months of May and August ($\beta_{ndvi}$) [67,72]. In addition to fixed effects, non-linear effects were included to control for strong autocorrelation among outbreak clusters ($\gamma_{clust}$, constructed covariate), distance to the nearest river ($\gamma_{river}$, Figure 1), and the N_*e*_ trajectory ($\gamma_{ne}$) as estimated during phylodynamic analysis (Section 2.4). These non-linear effects were modeled using a first-order random walk. However, $\gamma_{ne}$ was additionally constrained to be time-ordered. The sampling density estimated in the model’s first component was copied to the linear predictor (shared component, $W_{st}$) to help account for preferential sampling. Because the likelihoods differed between the two levels, the model-estimated parameter α was included to scale $W_{st}$. Environmental covariates incorporated in the model were selected based on prior VSNJV ecological analyses by Peters et al. [13], Palinski et al. [10], Burruss et al. [22], and Elias et al. [11]. Spatial statistical models were run using integrated nested Laplace approximation and the r-INLA package [73,74].

## 3. Results and Discussion

### 3.1. Phylogeography and Timeline

Phylogeographic analysis found a 2013 VSNJV isolate from Chihuahua, Mexico, to be the closest ancestral lineage to viruses sampled during the US 2014–15 outbreak cycle and rooted the time-resolved phylogeny to approximately November 2012 ± 10 weeks (Figure 2). During the intervening period between late 2012 and the first US detection on 23 May 2014 (index case, Texas), two distinct branching events occurred along the MCC phylogeny. The first bifurcation of interest occurred in approximately mid-2013. It produced two lineages: one line that would be ancestral to all US isolates sampled in 2014 and a second lineage that was a precursor to the 2015 outbreaks. The second branching event of note occurred between late 2013 and March 2014 and geographically divided what would become the 2014 line into geographically distinct “south” and “north” groups. The south group included the 2014 index case at the Texas–Mexico Border (May 23, Kinney County), whereas the north group was linked to outbreaks in Colorado [75].

### 3.2. Transmission Scenarios

Based on the phylogeny alone, major branch divergences before US disease detections suggest two possible scenarios:***Re-invasion:*** VSNJV lineage divergences occurred while the virus was geographically restricted to Mexico, followed by separate but recurrent US introductions in 2014 and 2015.***Overwintering:*** VSNJV lineage divergences occurred within the US following a single introduction event in late 2013. The 2013 incursion instigated the 2014 outbreaks, and the virus remained undetected through the US winter before re-emerging in 2015.

In addition to the re-invasion and overwintering scenarios inferred from the phylogeny, geographic uncertainty among samples collected near the US–Mexico boundary presented a third possibility: a diffuse border. In a diffuse border scenario, re-invasion and overwintering processes operate concurrently within southern portions of Texas, New Mexico, and Arizona, as these regions are ecologically continuous with Northern Mexico [76] and are known to experience comparable disease cohorts [77].

***Diffuse border:*** VSNJV exhibits transient persistence near the US–Mexico border, where introduced lineages persist during winter and are the source of US epidemics. The introduced lineages become locally extinct but are sequentially replaced through re-colonization by lines translocated from endemic regions in southern Mexico.

As evidence of possible border permeability, location uncertainty among VSNJV collected along the US–Mexico border resulted in the delineation of a “southern virus pool” (SVP) that was inclusive of the US state of Texas and the Mexican states of Chihuahua and Coahuila at the 80% credible interval (Figure 3). Further, the phylogeographic analysis showed that viruses collected in Colorado during both the 2014 (Figure 3a) and 2015 (Figure 3c) outbreaks separately descended from Texas lines from within this SVP.

Under the diffuse border scenario, the SVP is a fluctuating invasion front where VSNJV lineages are cyclically introduced from southern Mexico’s endemic regions and sequentially replaced. Following the introduction, epidemic viral lineages show transient persistence or short-term survival for a few years but are eventually subject to local extinction. As one lineage fades out or goes extinct, new, distinct VSNJV lineages are concurrently introduced through recolonization. This cycle of colonization, short-term persistence, and turnover supports the near-constant presence of some VSNJV variants, though the specific lineage dominating the landscape changes over time. Locations in southern Texas, New Mexico, and Arizona serve as transient corridors for VSNJV movement and are integral to the disease’s metapopulation structure. These areas facilitate and possibly filter lineage exchanges with Mexico and periodically act as conduits for seeding epidemic strains further north. Although SVP boundary uncertainty estimated in the current analysis does not allow us to confirm the diffuse border scenario, it leaves the possibility that endemic VSNJV lineages exhibit short-term persistence within the US southwest border region and are periodically the source of epidemic lines in the US.

During recent and historical conditions (see Burruss et al. [22] for VS under climate change), VS transmission from endemic regions in southern Mexico has not been sufficiently frequent or continuous enough to be established at higher latitudes; however, the more moderate climate found in the southwestern US may allow VSNJV to show short-term persistence [4,7,15] or to be sustained through horizontal, transovarial, venereal, and mechanical vector transmission [78,79,80]. Spatiotemporal heterogeneity among vector and host populations located between endemic Mexico and the US therefore plays a crucial role in shaping the fade-out and reemergence cycles observed in the VS system.

### 3.3. Epidemiological Dynamics

Spatial and temporal patterns observed during the 2014–15 VS outbreaks showed epidemic rather than endemic dynamics across the western US. Virus lineage dispersal rates estimated from genome sequences (Figure 4a) and distances between documented VS cases (Figure 4b) evidenced rapid dispersal at the beginning of the 2014–15 outbreak cycle as well as rate increases at the onset of each individual year. The average dispersal across the 2014–15 cycle was approximately 2 km/day; however, this rate was not constant and may have been 2–3 times faster in late 2013 and during the first halves of 2014 and 2015. It is noteworthy that dispersal rates derived from case surveillance showed an upswing in late 2014 but stalled before January 2015 (see arrow, Figure 4b). Opportunities for virus replication and mutation are proportional to the number of infected hosts; therefore, the temporarily variable dispersal velocities, heterogeneous branching rates, and overall geographic breadth of confirmed VS cases evidence an epidemic dynamic.

The connectivity or isolation among geographically dispersed populations determines their function as steppingstones, conduits, or barriers for virus translocation. Factors such as vector dispersal ability, mammalian host movements, fluctuating weather conditions, the quality of habitats, and the interactions between vectors and hosts collectively influence the likelihood of a pathogen persisting in a new location without succumbing to local extinction [81]. Situations of low disease prevalence often lead to stochastic fade-out, whereby the disease becomes locally extinct due to random variations in vector, host, or pathogen population sizes [82]. Historical phylogenetic data from outbreaks between 1981 and 2012 reveal a pattern wherein VSV lineages near the US–Mexico border in northern Mexico become locally extinct, only to be replaced by new lineages from endemic regions further south [83]. This pattern of stochastic fade-out followed by sequential replacement suggests that the conditions near the SVP, whether environmental, vectorial, or host-related, do not support sustained virus transmission or long-term viral persistence.

Although specific abundance thresholds determining pathogen persistence or extinction have been proposed for some disease systems, uncertain population estimates and the possibility of unknown reservoirs and host species complicate the picture [84,85,86]. Fade-outs are especially common during pathogen invasion or initial incursion, when a relatively low proportion of the susceptible population has been infected and the system is subject to random demographic (e.g., vector abundance changes), environmental (e.g., temperature and moisture variation), or genetic perturbations (e.g., virus selection or bottleneck) [87]. Even at low prevalence, fade-out is not assured, however, and pathogen incursion may be followed by secondary spread along intermittent or stuttering transmission chains that persist outside of endemic regions but below the epidemic thresholds that would enable detection [88,89].

Distinguishing between endemic and epidemic dynamics is central to understanding VS outbreaks in the US. Endemic and epidemic dynamics vary across several dimensions, including disease transmission rate, attributes of the susceptible population, and observable spatial and temporal patterns [90]. Except for rare endemic fade-out [81], disease transmission rates in endemic settings are generally stable through time. New infections occur relatively constantly, resulting in a steady-state equilibrium whereby case numbers remain mostly consistent and can be anticipated. In contrast, transmission rates increase rapidly during an epidemic, leading to surges in new infections and potentially exponential case growth. Host populations within a disease’s endemic area are likewise distinct from those found in epidemic regions: populations in endemic areas often show less susceptibility due to having developed immunity or resistance during past pathogen exposures. By comparison, populations in epidemic regions may be more susceptible to a disease due to relative immune naivety.

Several studies have also indicated that endemic and epidemic VSNJV strains may exhibit varying levels of virulence in specific hosts. For instance, a study by Velazquez-Salinas et al. [31] demonstrated that an epidemic strain induced a more robust immune response, higher fever, and increased lesions compared to the endemic strain. VSNJV strains displaying heightened virulence result in a more severe disease presentation that increases opportunities for host-to-host exchange, fomite contamination, and mechanical transfer to vectors, thereby facilitating faster and more extensive spread over greater distances compared to strains that cause mild or inconspicuous symptoms [23].

These epidemiological dynamics are reflected in the spatial and temporal patterns observed during an outbreak. Endemic diseases often exhibit stationary and predictable patterns within a geographic area. They are maintained in the absence of reintroduction or immigration. In contrast, pathogens in epidemic regions must first be introduced from an endemic region, only persist with rescue [91,92,93], and show less predictability as the result of rapid and widespread transmission. In part, a disease’s endemic or epidemic status determines its degree of predictability and affects how outbreak response is conducted.

### 3.4. Phylodynamics

The effective population size (N_*e*_) denotes the theoretical number of viruses in an idealized pool that would yield the same genetic variation and evolutionary patterns observed during the 2014–15 VS outbreak. Being tightly linked to pathogen diversity, the host population size, and the host infection number, N_*e*_ provides valuable insights into VSNJV transmission dynamics [47,48]. A higher N_*e*_ implies an expanded VSNJV pool, thereby indicating more host infections contributing to disease transmission. As estimated from the 2014–15 VSNJV genomes, N_*e*_ was positively correlated to VS incidence and displayed a temporal trend characteristic of an epidemic curve (Figure 5). That is, rather than indicative of an endemic steady-state equilibrium with values that fluctuate or cycle around a baseline number, N_*e*_ sharply increased in early 2014, peaked during the summer, and then abruptly declined by winter to form an “inter-epidemic trough” [92,94,95]. Following the winter bottleneck, the 2014 pattern mostly repeated in 2015, but summer N_*e*_ levels showed slightly more of a plateau that extended from early June through October 2015.

As a potential indicator of more efficient virus spread, N_*e*_ may foreshadow subsequent increases in disease incidence. When applied as part of spatiotemporal modeling, N_*e*_ was found to be an important indicator of disease intensity, with 95% credible intervals excluding zero during both the 2014 and 2015 outbreak seasons (Figure 4c). During 2014, the N_*e*_ coefficients displayed negative polarity, signifying relative case dilution due to the rapid increase in the total geographic area encompassing all VS cases. Recall that intensity refers to the number of premise-level disease cases estimated on a per-unit-area basis (cases/5 km^2^). Thus, intensity values can decrease due to reduced incidence or increased area. Although the total outbreak area continued to grow in 2015, the expansion rate was slightly lower than in 2014 and ceased by mid-year (Figure 4b), resulting in positive N_*e*_ coefficient polarity during the outbreak cycle’s second year. The overall temporal trajectory of N_*e*_ resembles transmission dynamics in which VSNJV either evolved increased virulence or was exchanged among spatially disjunct vector–host meta-populations. This interpretation underscores the interconnectedness of N_*e*_ with VS spatial epidemiology, providing a fuller understanding of how pathogen dynamics unfold over time within a larger geographic context.

Pathogen movement or migration between spatially separated meta-populations shapes the persistence patterns seen at regional levels [95]. The inter-epidemic troughs associated with incidence curves (or N_*e*_) signify decreases in pathogen prevalence and genetic diversity following local extinctions. In time, however, pathogens may be reintroduced (rescued, sensu Brown and Kodric-Brown [91]) to populations that have lost immunity or have been replenished with newly susceptible individuals (e.g., births and/or immigration) [95]. Through these spatial meta-population dynamics, a pathogen may become recurrently extinct at any given location but still be maintained in the larger geographic region, where it serves as a source for reintroduction [96]. At the local scale, the probability of epidemic fade-out decreases with increasing vector or host population sizes; thus, VSNJV N_*e*_ variation (Figure 5) may have been as attributable to changes in vector population abundance as it was ascribable to vertebrate host availability and incidence rate. In particular, decreased VSNJV N_*e*_ during the winter bottleneck between 2014 and 2015 may mechanistically resulted from an insect vector abiotic limitation.

VSNJV detections and reported VS cases declined sharply between the fall of 2014 and the spring of 2015: a trend likely driven by diminished vector abundance during the colder US winter season. This pattern underscores the critical influence of temperature-dependent vector dynamics on disease transmission [4,7,10,13]. Historical data from Mexico between 1981 and 2012 evaluated by Navarro López et al. [83] provide a comparative backdrop, illustrating how endemic regions in Mexico, characterized by subtropical climates, exhibit different seasonal dynamics compared to the temperate regions that were affected by the 2014–15 VSNJV outbreaks. While temperate regions experience four distinct seasons with significant temperature fluctuations that influence vector activity and disease transmission [97], subtropical endemic areas in Mexico undergo primarily wet and dry seasons [98,99], leading to a more stable climate that is conducive to continuous vector presence. This distinction is pivotal for understanding VSNJV transmission dynamics and suggests that the broader temperature ranges in temperate regions may limit insect abundance, instigating local pathogen extinction during winter. Conversely, a more moderate climate may be a prerequisite for virus persistence, as observed in VSNJV endemic areas. Even though broader temperature ranges potentially limited insect abundance during the US 2014–2015 winter, VSNJV may still have persisted in the southwestern US [4,7,15,79] and provided rescue or sourced re-introductions when insect populations rebounded the following spring.

### 3.5. Transmission Network

Like the phylogeographic analysis, transmission network reconstruction suggested that VSNJV was first introduced to Texas locations within the SVP sometime in late 2013 or early 2014 (Figure 6). From the SVP, the virus spread eastward in Texas and then northward to Colorado later in the year (Figure 6, left). Transmission dynamics revealed that unknown hosts were likely transmitting what would become the 2014 incursion lineage by January 2014—several months before the US index case at Kinney County, Texas, on 23 May 2014 [75]. Although the geographic locations of the inferred but unsampled hosts between Chihuahua and Texas cannot be pinpointed precisely, VS cases within the SVP were documented in Mexico’s Sonora and Chihuahua states in late 2013 [100]. VS cases in northern Mexico have been observed to precede US epidemics during other outbreak years [1].

The phylogeographic (Figure 3a,b) and transmission network (Figure 6) analyses concurred that by mid-2014, VSNJV had spread from Texas to Colorado. Texas VS incidence then decreased sharply during the late summer of 2014, but cases continued in Colorado through November, ultimately seeding a Nebraska outbreak in early November 2014. Interestingly, the next VS cases to be documented occurred in mid-December 2014 and early February 2015 within the Mexico-bordering county of Santa Cruz, Arizona (confirmed by APHIS [75] on 6 January 2015). Genetic samples were not collected from either case; therefore, sequences were unavailable for phylogenetic analysis. However, transmission network analysis did infer the occurrence of a contemporaneous VSNJV host with an estimated December 2014 virus infection date.

Transmission network analysis indicated that the VSNJV lineage leading to the 2015 outbreaks exhibited greater parity to sequenced 2014 Texas viruses than to hosts along the transmission chain linking the Chihuahua isolate to the Texas index case. Therefore, the most probable transmission chain moved the virus from Mexico to Texas via unknown hosts in late 2013 before being propagated from the index case to other US localities in 2014 and 2015 (Figure 6). Concurrent with the 2014 outbreak, the lineage that emerged from the SVP in Texas continued evolving among unsampled hosts before instigating outbreaks in Arizona, South Dakota, Colorado, and other Texas locations during 2015. The most recent unknown host in the transmission chain between the 2014 Texas viruses and the 2015 Arizona outbreaks was estimated to have been infected during the first week of December 2014—only a week before infections were confirmed in Santa Cruz, Arizona. Given that the unsampled host was inferred to have evolved from 2014 Texas viruses and was also ancestral to 2015 viruses in Texas, it is reasonable to assume the host to have overwintered in Texas or elsewhere in close SVP proximity.

Paraphrasing Robert May [101], the strength of a model lies in its capacity to impose clarity and precision on speculative ideas, fostering a valuable comparison between fundamental assumptions and observed facts. Similarly, phylogenies are only models for illustrating evolutionary relationships and may not accurately translate biological reality, especially with incomplete sampling. Though they bring precision to hypotheses, their inherent abstraction must be acknowledged [102,103]. Biases introduced by imperfect detection, sampling procedures, and analysis assumptions can distort epidemiological interpretation [104].

As a case in point, phylogeographic analysis indicated that VSNJV sequences collected from central Arizona in mid-May 2015 were most probably descendants of Utah isolates collected in late April. However, closer examination of phylogenetic transition probabilities revealed that virus translocation from western Utah to Arizona was only 10% more probable than transfer in the opposite direction: Arizona to Utah. Transition matrices also reflected a low probability (<10%) that the VSNJV detected in western Utah had been seeded from eastern Utah: the most geographically proximal virus source (Figure 3c). Further complicating the picture, APHIS reported four other VS cases preceding the west Utah case: two from central Arizona and two from southern Utah [75]. Although lacking genetic samples, the relative timing and geographic positioning of the Arizona and south Utah cases, in combination with infection chains identified by the transmission network and geographic transition probabilities, suggested that a northward transfer (Arizona to Utah) was a more parsimonious explanation of observed epidemiological patterns. On reviewing these conclusions with APHIS staff, it was learned that the April 2015 detections in Utah were confirmed through epidemiological investigation to have resulted from unauthorized horse movement from Arizona to Utah. In addition to demonstrating how incorrect conclusions may be drawn from phylogeographic analyses with imperfect sampling, this information also provides strong evidence of a VSV incursion near the Mexico-bordering county of Santa Cruz, Arizona, between December 2014 and February 2015.

Despite the phylogeographic and transmission network analyses being in broad agreement, sample uncertainty and methodological differences leave the exact number, timing, and order of several other transmission events ambiguous. For example, it is unclear if the 2015 virus transfer from the SVP to Colorado was direct or via a stopover in Reeves County, Texas (Figure 3c), and if 2015 South Dakota outbreaks were an extension of the SVP–Colorado chain or instead resulted from virus introduction from the SVP directly (Figure 3d).

### 3.6. Vector–Host Interactions and Environment

VS outbreak intensity was spatially and temporally heterogeneous throughout the 2014–15 epidemic (Figure 3bottom). After accounting for preferential and incomplete sampling, the model estimated VS intensity ranged from 0 (no cases) to 10 cases/5 km^2^, with the most intense outbreak rates identified for eastern Colorado during 2014 and 2015. Disease clusters or “hotspots” were also identified for south-central Texas in 2014 and north-central Utah, east Texas, and central Arizona in 2015. In epidemiology, an outbreak cluster refers to an unusually high number of related disease cases within a specific geographic area or time frame; such localized concentrations may indicate increased transmission rates or a common origin or exposure source [7]. Spatiotemporal modeling indicated that VS outbreaks clustered (see $\gamma_{clust}$, Section 2.5) at a distance of approximately 10 km (Figure 7a), meaning that elevated case intensity was observed within a 10 km radius of an identified VS case, but VS risk quickly decreased beyond this distance threshold. The 10 km threshold estimated in this study is consistent with minimum quarantine and control area recommendations for foot-and-mouth disease [105,106] and emergency responses for other foreign animal diseases [107].

Geographic proximity to Class 1 and 2 Rivers (see, $\gamma_{river}$, Section 2.5) was a statistically important indicator of outbreak intensity (Figure 7b) and suggested that VS case numbers sharply decreased beyond approximately 20 km distance from flowing water. Flowing water proximity can influence VS transmission through effects on vector insect reproduction and host animal behavior. As examples, black flies (e.g., *Simulium vittatum*) are recognized VSNJV vectors that require flowing water for reproduction [108], exposure risk may increase due to livestock congregating near watering sources [109], and seasonal streamflow patterns have been correlated to past VS detection [11,110]. The 20 km threshold identified through the current analysis falls within black fly dispersal ranges, which exhibit an average dispersal from flowing water of approximately 10 km and an upper limit of around 35 km [111]. Using artificial intelligence and machine learning methods, Peters et al. [13] also found stream proximity within 17 km associated with higher VS cases. However, Peters et al. [13] also detected a dynamic in which stream proximity may hold greater importance during an initial incursion year than in the following expansion year.

Although Peters et al. [13] identified distinct water proximity effects between incursion and expansion years, the current study found outbreak clustering radii and river proximity distances to be mostly consistent between 2014 and 2015. VS clustering distances were slightly wider in 2015 than in 2014. However, credible intervals for the two years substantially overlapped (Figure 7a), indicating inclusive coefficient ranges. Likewise, the relationship between river proximity and outbreak intensity was similar in both years, albeit with the 2014 outbreak showing a somewhat greater intensity within 20 km of flowing water (Figure 7b). Despite small coefficient sizes, the between-year differences shown by clustering and river proximity were not contradictory to Peters et al. [13] and aligned with an incursion–expansion hypothesis. Unlike stream proximity, between-year differences were noted among other assessed environmental variables, particularly those associated with moisture and water availability.

Spatiotemporal modeling identified statistically important between-year differences among several environmental variables. Among these variables, the EDDI dryness index was found to be positively related to VS intensity in 2014 but negatively associated in 2015, topographical wetness had a negative effect in 2014 but a positive influence in 2015, temperature was important in 2015 but not in 2014, and seasonality was significant in 2014 but not 2015. Soil carbon was positively associated with case intensity in 2014 but not in 2015 (Figure 8). A couple of caveats are necessary when interpreting these results. First, though our statistical approach enabled correlations and associations between case intensity and environmental factors to be quantified, mechanistic interpretations should be avoided as the method did not include formal causal analysis [112,113]. Secondly, apart from the EDDI [66] and NDVI [72] variables aggregated to the year-specific season, all other environmental variables reflect static, long-term conditions across the study area. Specifically, temperature and seasonality [69] represent long-term climate averages, whereas the topographic wetness index [68], soil organic carbon [70], shrub-occupied land cover [71], and elevation [68] reflect prevailing landscape characteristics related to edaphic conditions and physiographic positions.

Given the above considerations, the environmental analysis showed that 2014 VS cases correlated with anomalously arid conditions (as measured by the EDVI) and that higher case numbers were experienced in areas that did not typically retain water (Wetness). Locations with increased case intensity in 2014 typically included low-lying vegetation (Shrub Landcover) with organic soils (Soil Carbon) and exhibited significant seasonal temperature fluctuations (Seasonality) (Figure 8). By comparison, 2015 cases were associated with greater moisture (EDDI), areas of accumulated water (Wetness), and locations with above-average minimum temperatures (Temperature) relative to the study-area-wide average. Elevation and NDVI variables were consistent between years, likely due to VS cases spanning a wide altitudinal range and the entirety of the summer season in both 2014 and 2015.

VSNJV transmission dynamics are not only driven by evolutionary processes such as pathogenicity, host competence, and adaptation during host-switching events [10] but are also significantly influenced by the environmental conditions that affect vector abundance and feeding preferences [19,20,21]. Although the complete mechanisms of VSNJV transmission remain elusive, isolation of the virus from various arthropods implicates a complex vector community role in disease propagation [4,23]. Consequently, each vector specie’s contribution toward disease spread may fluctuate independently of other species over the temporal and spatial dimensions.

This complexity is exemplified by vector taxa such as biting midges (*Culicoides* spp.) and black flies (*Simulium* spp.), which are illustrative of distinct ecologies and behavior in the VS system. Biting midges exhibit a predilection for warm, humid environments near organic-rich soils, vegetation, and stationary water bodies [114]. On the other hand, black flies are more dependent on flowing water for oviposition, hatching, and larval development and tend to be less drought tolerant than biting midges [8,13]. Feeding behavior may also show genera-specific differences, with probing behaviors, feeding site preferences, co-feeding proclivity, and host specificity varying between groups [4].

The interplay between vector-specific traits and environmental variation plays a pivotal part in shaping VS epidemiology. Environmental changes across different locales and over multiple temporal scales impact the distribution, density, survival rate, and biting activity of vector species, which, in turn, can either enhance or diminish vectorial capacity [4,11,115,116,117]. In combination with VS clinical manifestation, environmental variation can likewise modify vertebrate host behavior to further influence disease spread. For example, limited watering sources can cause animals to aggregate during a drought, increasing the likelihood of host-to-host contact even as animals exhibit heightened viral shedding due to severe symptoms [4].

Understanding these complex vector–environment interactions informs the development of effective VS management strategies. Tailoring policies to accommodate the varied behaviors of vector populations—accounting for their locality-specific and time-sensitive roles in disease dynamics—could facilitate targeted interventions that reduce pathogen prevalence [27,118]. By focusing on the most influential vector species within particular environments and appropriate time windows, it is possible to better control and mitigate outbreak severity and frequency, improving VS management across affected regions [13,118].

## 4. Conclusions and Summary

In our study, we used an innovative analytical pipeline to understand the epidemiological patterns of the Vesicular Stomatitis New Jersey virus (VSNJV) during the 2014–15 epizootic cycle in the United States (US). We conducted an integrative analysis that combined VSNJV genomics data, confirmed vesicular stomatitis (VS) disease cases from surveillance, and varied environmental factors. By applying spatial epidemiology as an interpretive scaffolding, we assessed the continuity between VS cases reported in 2014 and those in 2015. Our aim was to determine whether multiyear outbreak patterns were more consistent with US viral overwintering or repeated virus introductions from endemic Mexico.

The spatial and temporal dynamics characterizing the 2014–15 VS outbreak were indicative of an epidemic phenomenon rather than the stable transmission associated with endemicity. Despite the inherent uncertainties due to incomplete virus sampling and multi-scaled spatial processes, our findings support the occurrence of multiple independent re-invasion events concurrent with potential viral overwintering between sequential seasons. Further, VSNJV invasions and overwintering sites likely emanated from a geographically defined southern virus pool (SVP) located at the US–Mexico border interface.

We observed a contraction in genetic diversity during the 2014–15 inter-epidemic period, which could be attributed to overwintering or subsequent re-introductions of the virus. However, given the continuous presence of VSNJV within the SVP and strong environmental determinants impacting vector population abundance during the winter season, we inferred that the overwintering hypothesis had greater mechanistic credibility for explaining the genetic bottleneck. Changes in genetic diversity, as measured by the VSNJV effective population size (N_*e*_), were found to precede and strongly correlate with VS incidence, further demonstrating an epidemic pattern. These findings support the concept of meta-population dynamics functioning among spatially dispersed vector populations between Mexico’s endemic regions and episodically affected areas in the US.

Understanding VS outbreaks necessitates unraveling the interplay between vector species preferences and behavior, coupled with environmental spatial and temporal heterogeneity. Environmental factors are central in shaping, sustaining, and modifying suitable conditions for vector populations. Distinct climatic and edaphic settings can favor one vector species over another, influencing vector–host interaction and modifying disease transmission risk across multiple scales. These interlocking factors highlight the imperative for adaptive, location-specific strategies to respond to and manage VS outbreaks.

Effective VS response and management requires a comprehensive approach informed by VS evolution and ecology. This entails considering the evolutionary processes that drive virus diversity and transmission efficiency as well as the ecological processes that shape vector population dynamics, feeding behavior, and interactions with livestock. In the context of epidemics, a heightened and decisive response is imperative and involves rapid and aggressive measures to effectively curtail disease spread. These measures encompass actions such as quarantines, widespread testing, movement restrictions, and contact tracing aimed at preventing the escalation of outbreaks on a large scale. Furthermore, implementing tailored strategies that focus on monitoring virus diversity and managing key vector species at specific locations and times can significantly reduce pathogen prevalence, providing robust and effective countermeasures against VS epidemics.

Addressing the broader impacts of VS outbreaks extends beyond disease containment. It involves financial support for affected industries, transparent communication strategies, and fostering international cooperation, particularly with partners like Mexico. It also underscores the need for sustained research investment for expanded field sampling, enhanced laboratory capacity, and the development of integrative modeling techniques to refine our understanding of VSNJV spatial epidemiology.

## Figures and Tables

**Figure 1 viruses-16-01118-f001:**
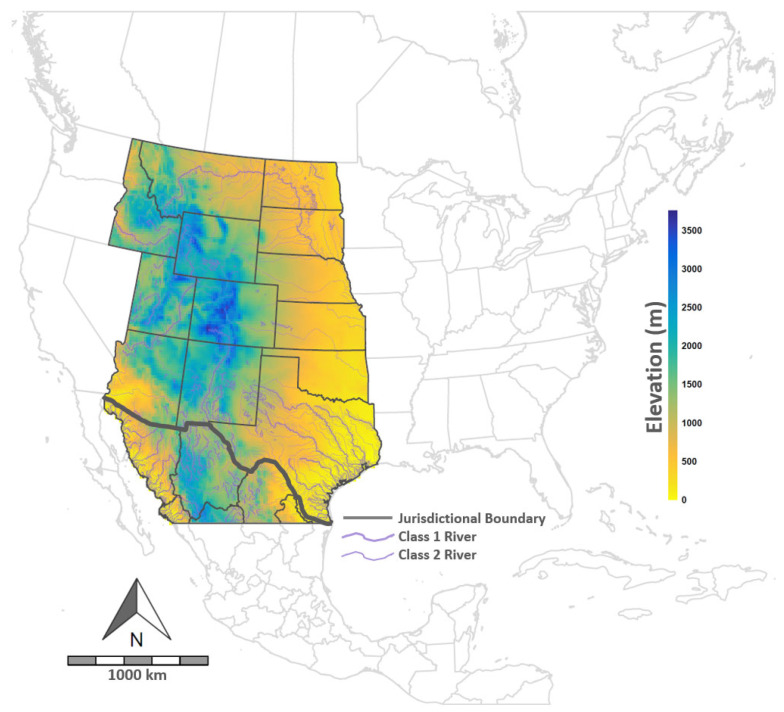
Study domain. Map is color-coded according to the legend at right to approximate elevations in the study area and to depict Class 1 and 2 rivers that were used in some analyses. Project boundaries were selected based on VS outbreak distributions, as illustrated in Figure 2.

**Figure 2 viruses-16-01118-f002:**
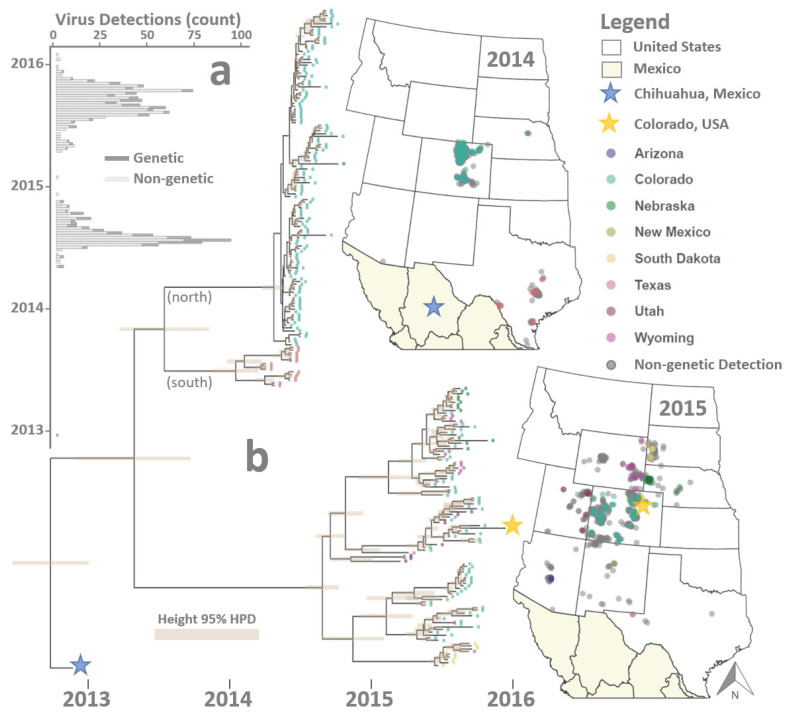
Observation data. Figure illustrates sample times and locations for 1246 VS disease case detections and 241 genetic samples. Panel (**a**) provides a bar plot of sample counts (horizontal axis) by date (vertical axis), with genetic samples differentiated from non-genetic case detections by color. Panel (**b**) displays a time-resolved phylogenetic tree of virus sequences with tips color-coded and shaped to correspond to mapped collection locations in 2013–2016. Maps depict sample geographic locations, with star-shaped points indicating the first (blue) and last (yellow) samples in the analysis.

**Figure 3 viruses-16-01118-f003:**
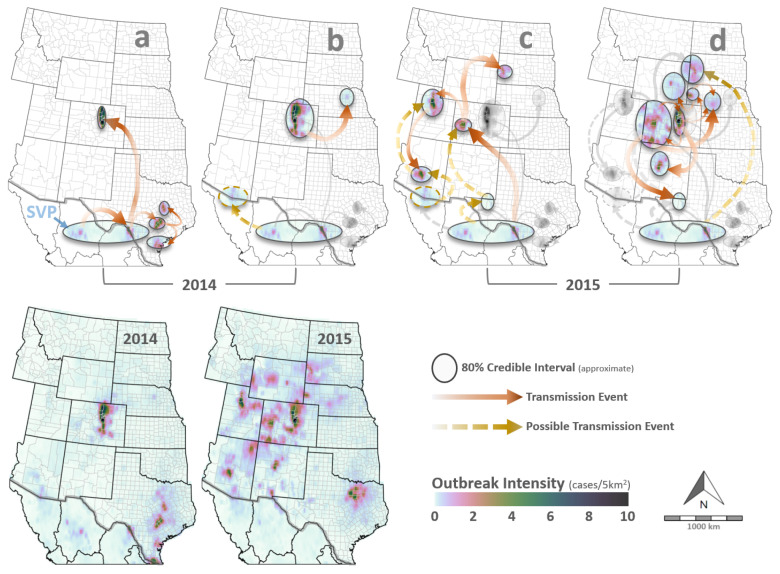
VS phylogeography. Maps summarize results from virus phylogeographic analysis and disease intensity modeling. The top row maps spatiotemporal relationships among genomes sampled from 2013 to 30 June 2014 (Panel (**a**)), the period of 1 July–31 December 2014 (Panel (**b**)), 1 January–30 June 2015 (Panel (**c**)), and 1 July–31 December 2015 (Panel (**d**)). Circular shapes overlaid in the top row approximate the location uncertainty (80% credible intervals), with arrows signifying potential dispersal events between locations. SVP stands for “Southern Virus Pool” and highlights the location uncertainty along the US–Mexico border. Red–brown arrows represent dispersal events estimated directly by the phylogeographic analysis, whereas gold–yellow arrows signify potential transmission events inferred from transmission models (Figure 6) and descriptive epidemiology. The bottom row panels display the estimated outbreak intensity (cases/5 km^2^) for 2014 (**left**) and 2015 (**right**) and are color-coded according to the legend at the bottom right.

**Figure 4 viruses-16-01118-f004:**
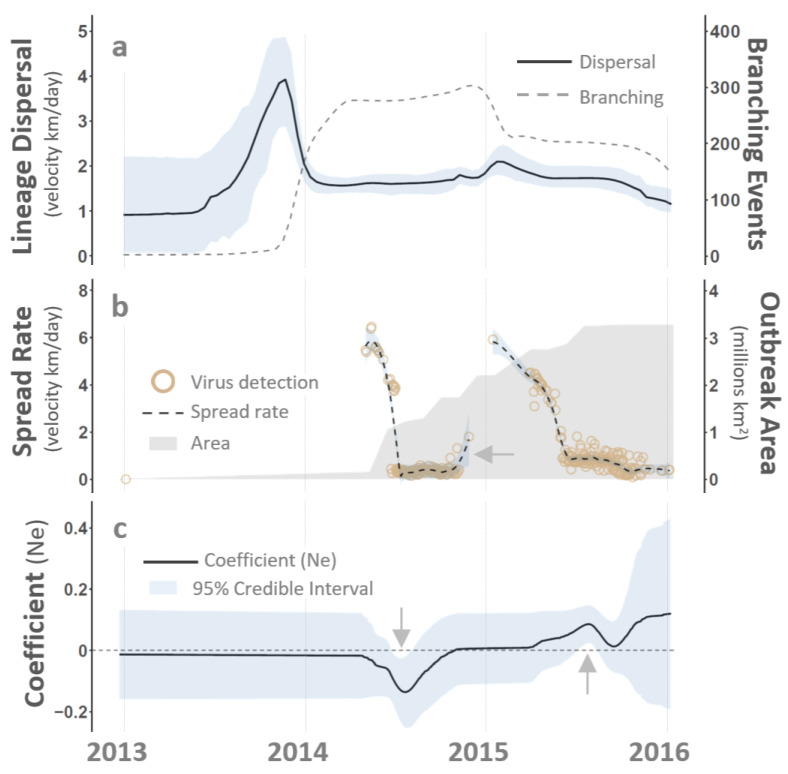
Virus lineage dispersal and virus population dynamics. The figure summarizes the lineage dispersal rates relative to phylogenetic branching (Panel (**a**)), the geographic spread rate among documented VS cases (Panel (**b**)), and the influence of the virus’s effective population size (N_*e*_) on outbreak intensity (Panel (**c**)) as functions of time (horizontal axis at the bottom). Panel (**a**) illustrates increased phylogenetic branching (dashed line) immediately following virus introduction (smooth line, maximum). Panel (**b**) shows peak VS spread rates (dashed line, approximately 6 km/h) at the onset of the 2014 and 2015 seasons and relative to the total geographic area affected by disease (gray shading). In Panel (**b**), the arrow emphasizes an uptick in the spread rate in late 2014. Panel (**c**) displays the time-varying coefficient (smooth line) for the relationship between N_*e*_ and the outbreak intensity. In Panel (**c**), the arrows highlight the 95% credible interval departures from the average outbreak intensity (scaled to zero, horizontal line), indicating periods of statistical importance. Raw N_*e*_ estimates from phylodynamic analysis are given in Figure 5. See the main text for a detailed description.

**Figure 5 viruses-16-01118-f005:**
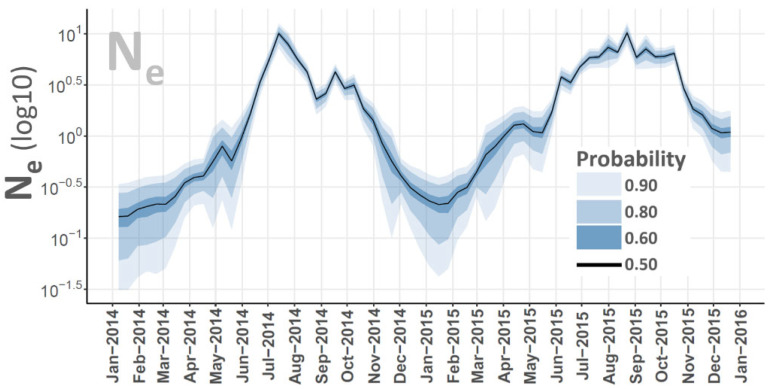
Effective population size. VSNJV effective population size (N_*e*_). The horizontal axis lists dates between January 2014 and January 2016, and the vertical axis displays scaled N_*e*_. The inset legend at the lower right shows credible intervals, with the solid black line representing the median point estimate. The estimated influence of N_*e*_ on the outbreak intensity is illustrated in Figure 4c.

**Figure 6 viruses-16-01118-f006:**
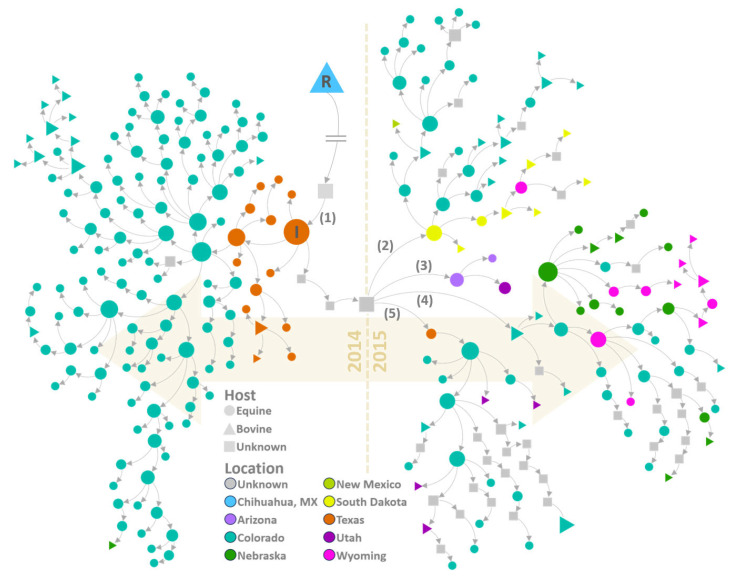
VSNJV transmission tree. Figure illustrates the VSNJV transmission network derived from the time-resolved VSNJV phylogeny rooted (R) using a sample from Chihuahua, Mexico. Arrows indicate probable transmission events between hosts, which are shown as nodes. Tree nodes represent host species and are color-coded according to the legend at the bottom center to indicate sample location. Node sizes approximate the relative number of infections resulting from each host; larger nodes indicate more individuals were infected. The US index case from 23 May 2014 (Kinney County, Texas) is annotated with the letter “I”. Gray nodes (unknown hosts) are probable transmission events corresponding to unsampled virus genomes. Except for the 2013 Mexico sample (R), the dashed vertical line partitions 2014 transmission events (**left** side) from those from 2015 (**right** side) and intersects an unsampled host estimated to have been infected in mid-December 2014. Parenthetical numbers highlight the (1) transmission event leading to the 2014 US index case and possible 2015 transmission events from an unsampled host to (2) South Dakota, (3) Arizona, (4) Colorado, and (5) Reeves County, Texas.

**Figure 7 viruses-16-01118-f007:**
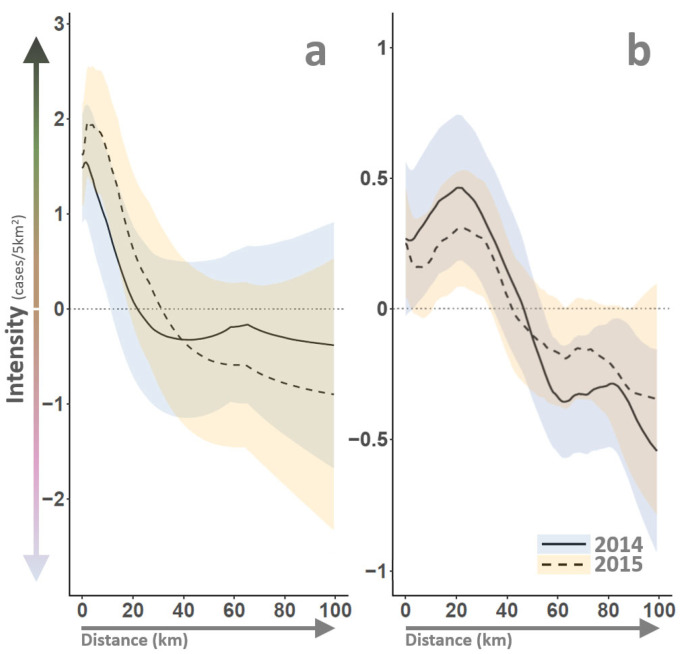
Outbreak cluster and stream distances. VS outbreak cluster distances (Panel (**a**)) and river proximity (Panel (**b**)) are illustrated, with horizontal axes indicating geographic distance (km) and vertical axes reporting relative change in outbreak intensity (cases/5 km^2^). Vertical axes have been scaled and centered to have a mean of zero (horizontal dotted line at panel centers). Panel (**a**) indicates that VS cases “clustered” or exhibited elevated intensity within an approximate 10 km radius. Panel (**b**) highlights that areas within about 20 km of a Class 1 or 2 river (see Figure 1) displayed elevated VS intensity. Outbreak years are distinguished by line type and include shaded 95% credible intervals as depicted in the legend at bottom right. Note that line portions excluding zero (dotted horizontal line) suggest statistically important influence.

**Figure 8 viruses-16-01118-f008:**
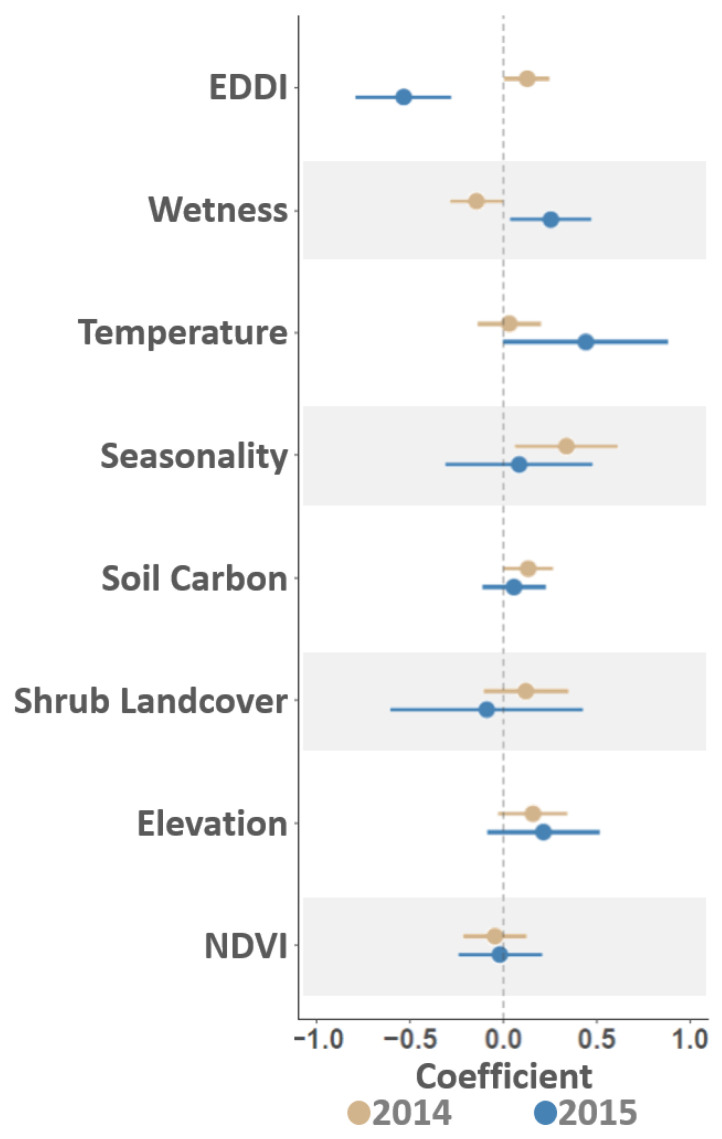
Estimated coefficients. Figure compares associations between VS outbreak intensity (cases/km^2^) and several environmental variables. Evaluated variables as listed on the vertical axis included drought conditions (EDDI), a topographical Wetness index, average minimum Temperature, a Seasonality index quantifying seasonal temperature variation, Soil Carbon, the proportion of Shrub Landcover within 5 km^2^, Elevation, and a Normalized Difference Vegetation Index (NDVI). Note that even after accounting for spatial correlation, temporal correlation, and preferential sampling biases, VS associations to drought conditions and environmental moisture varied between the 2014 and 2015 outbreak years. The contrasting relationships for these water-based influences may be suggestive of arthropod host switching, weather limitations, pathogen evolution, or other epidemiological factors contributing to incursion and expansion dynamics in the VS disease system.

## Data Availability

The GenBank and Sequence Read Archive (SRA) accession numbers for the genetic sequences used in this study are provided in the Appendix A. The code and simulated data to construct and run the spatial statistical analyses presented in this manuscript are available at https://geoepi.github.io/vs-epizootics and have been archived (DOI:10.17605/OSF.IO/GHZFQ) on the Open Science Framework at https://osf.io/ghzfq/. The disease incidence and genetic data utilized in this study include geographic coordinates and other personally identifiable information that discloses the locations of private and commercial livestock facilities. Safeguarding the confidentiality and privacy of this sensitive information is crucial to complying with ethical and legal standards; therefore, exact geographic coordinates have been removed from all data. Researchers interested in accessing the pertinent data for replication or additional investigation may contact the corresponding author to discuss potential avenues for data access while upholding privacy and confidentiality standards.

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
