# Peer review of "Interrogating Genomes and Geography to Unravel Multiyear Vesicular Stomatitis Epizootics"

_viruses, 2024, doi:10.3390/v16071118_

Round 1

Reviewer 1 Report

Comments and Suggestions for Authors

Overall, this is a very solid submission containing potentially significant findings to the field of VSV epidemiology and disease mitigation.  I have a few small and one major request to improve this submission.  My major concern is the lack of public availability of the viral sequences described in this submission.  I acknowledge that lines 664-670 mention issues related to confidentiality, privacy and avenues of potentially sharing some data however I find this lacking.  Before I can recommend this submission for publication, I would like to see the sequences generated in this submission, stripped of precise geo-coordinates, publicly available in GenBank or hear more substantial arguments why the sequences, stripped of precise geo-coordinates, cannot be added to the GenBank database.        

              Aside from the major concern regarding sequence availability, I have only two minor requests for this submission, both in regards to the Materials and Methods.  Would the authors kindly:

·       Clarify the material sequenced.  Were these original field samples or passaged in cell culture?  If passaged, what cell line and how many passages? (submission lines 101-106)

·       Provide the accession numbers of the two references used in read mapping.  (line128)

In summary, I feel this submission is very well-crafted, well written and thorough.  Aside from a major concern regarding viral sequence availability, I only note two minor additions for the Methods.

Author Response

REVIEWER 1

Comment 1: Overall, this is a very solid submission containing potentially significant findings to the field of VSV epidemiology and disease mitigation.  I have a few small and one major request to improve this submission. 

Response 1: We thank the reviewer for the time taken to evaluate the manuscript and the comment regarding the manuscript providing potentially significant findings.

Comment 2: My major concern is the lack of public availability of the viral sequences described in this submission.  I acknowledge that lines 664-670 mention issues related to confidentiality, privacy and avenues of potentially sharing some data however I find this lacking.  Before I can recommend this submission for publication, I would like to see the sequences generated in this submission, stripped of precise geo-coordinates, publicly available in GenBank or hear more substantial arguments why the sequences, stripped of precise geo-coordinates, cannot be added to the GenBank database.   

Response 2: We thank the reviewer for emphasizing the importance of public data access.  The sequences, stripped of exact geographic coordinates, have been submitted to GenBank and are currently pending public release.  A supplementary file with this revised manuscript includes the GenBank accession number, Sequence Read Archive (SRA) accession number, sample collection date, sampled species (bovine or equine), and the collection location to the level of U.S. County.  Accordingly, the Data Availability Statement has been updated to reflect sequence information (Lines 668-669).

Comment 3: Aside from the major concern regarding sequence availability, I have only two minor requests for this submission, both in regards to the Materials and Methods.  Would the authors kindly: Clarify the material sequenced.  Were these original field samples or passaged in cell culture?  If passaged, what cell line and how many passages? (submission lines 101-106)

Response 3: This section has been revised to specify that cell cultures were used and cell lines differed based on where the material was obtained.  Depending on the source, samples were received as passage 1 or 2 supernatant recovered from Vero, porcine kidney (IBRS-2), Lamb Kidney (LK), or  Baby Hamster Kidney (BHK) cell lines.  Please see Lines 104-110 for details.

Comment 4: Provide the accession numbers of the two references used in read mapping.  (line128)

Response 4: Thank you for identifying the need to cite the reference sequences properly; the GenBank accession numbers and associated publication citations for each sequence have been added to the methods section at Lines 132-134.

Comment 5: In summary, I feel this submission is very well-crafted, well written and thorough.  Aside from a major concern regarding viral sequence availability, I only note two minor additions for the Methods.

Response 5: Again, we thank the reviewer for the comments and suggestions for improvement.

Reviewer 2 Report

Comments and Suggestions for Authors

This manuscript presents a genomic analysis combined with geographical and environmental data on the outbreak of vesicular stomatitis New Jersey virus (VSNJV) during the 2014-15 epizootic in the US. VSNJV is endemic in Mexico and Central America, and causes periodic outbreaks in the US, resulting in significant economic losses in domestic livestock, and presenting clinical manifestations in cattle and swine indistinguishable from that of foot-and-mouth disease. The epidemiology of VSNJV is complicated by transmission by multiple arthropod vectors, and many questions remain about the environmental and other factors leading to these outbreaks. The data in the current manuscript are consistent with earlier data supporting an “incursion-expansion” model, in which virus introductions from Mexico are followed by overwintering, geographic expansion, and eventual local extinction in the US. The current manuscript adds to this body by the impressive layers of detail resulting from the high quality and quantity of the data and modern analytical methods. The results are consistent with a geographically defined Southern Virus Pool at the US-Mexico interface as the source of VSNJV invasions and overwintering sites. Genomic analysis indicated an increase in virus diversity before a rise in case numbers and a pronounced reduction in virus diversity during the winter season, indicative of a genetic bottleneck and a significant narrowing of virus variation between the summer outbreak seasons. Environmental data were consistent with different vector interactions during the expansion phase in 2014 versus 2015.

              Overall the manuscript is very well-written. The explanations of complicated phylogenic and epidemiological data are clearly presented, and the limitations on the conclusions to be drawn are stated. Whereas this reviewer is not familiar with all of the analytical methods described, they appear to be similar to those used with other viruses, leading to the most probable models that account for the data. The only minor criticism I had was in the section “Epidemiological Dynamics”, in which a lengthy discussion is followed by the last paragraph describing the results of the present study. This section would perhaps be clearer if it followed the usual order of stating the results first followed by a discussion that puts them in context.

Author Response

REVIEWER 2

Comment 1: This manuscript presents a genomic analysis combined with geographical and environmental data on the outbreak of vesicular stomatitis New Jersey virus (VSNJV) during the 2014-15 epizootic in the US. VSNJV is endemic in Mexico and Central America, and causes periodic outbreaks in the US, resulting in significant economic losses in domestic livestock, and presenting clinical manifestations in cattle and swine indistinguishable from that of foot-and-mouth disease. The epidemiology of VSNJV is complicated by transmission by multiple arthropod vectors, and many questions remain about the environmental and other factors leading to these outbreaks. The data in the current manuscript are consistent with earlier data supporting an “incursion-expansion” model, in which virus introductions from Mexico are followed by overwintering, geographic expansion, and eventual local extinction in the US. The current manuscript adds to this body by the impressive layers of detail resulting from the high quality and quantity of the data and modern analytical methods. The results are consistent with a geographically defined Southern Virus Pool at the US-Mexico interface as the source of VSNJV invasions and overwintering sites. Genomic analysis indicated an increase in virus diversity before a rise in case numbers and a pronounced reduction in virus diversity during the winter season, indicative of a genetic bottleneck and a significant narrowing of virus variation between the summer outbreak seasons. Environmental data were consistent with different vector interactions during the expansion phase in 2014 versus 2015.

              Overall the manuscript is very well-written. The explanations of complicated phylogenic and epidemiological data are clearly presented, and the limitations on the conclusions to be drawn are stated. Whereas this reviewer is not familiar with all of the analytical methods described, they appear to be similar to those used with other viruses, leading to the most probable models that account for the data.

Response 1: We thank the reviewer for the time provided to review the manuscript and the comments indicating the clarity and thoroughness of the manuscript.

Comment 2: The only minor criticism I had was in the section “Epidemiological Dynamics”, in which a lengthy discussion is followed by the last paragraph describing the results of the present study. This section would perhaps be clearer if it followed the usual order of stating the results first followed by a discussion that puts them in context.

Response 2: We concur with the reviewer regarding the arrangement and order of text within the “Epidemiological Dynamics” subsection and have revised the manuscript to relocate the description of model results to the first paragraph of the subsection.  Please see Lines 316- 327 in the revised manuscript.

Reviewer 3 Report

Comments and Suggestions for Authors

Comments to authors

In this manuscript entitled “Interrogating Genomes and Geography to Unravel Multiyear Vesicular Stomatitis Epizootics”, J M Humphreys et al presented a comprehensive study to elucidate the spatial epidemiological patterns of the Vesicular Stomatitis New Jersey virus during the 2014-15 epizootic cycle in the United States. They found that a geographically defined Virus Pool at the US-Mexico interface was the source of VSNJV invasions and overwintering sites. Generally, the data sources are credible, the data analysis in this study is coherent, the inferences are proper and the results are meaningful and explicable.

The main deficiency is that the viral isolates from Mexico contain only 1 sample (n=1). Due to the high mutation in single-strand RNA viruses, too few samples may lack typicality.

In Figure 7b, the VS case numbers sharply decreased beyond approximately 20 km distance from the river. Except for the vector insect reproduction and host animal behavior, does the amount of host animals also affect the VS case numbers? After all, most pastures are distributed along the river for an abundant water supply.

The following minor issues need consideration.

1. In Figure 1, the lines indicating River Class 1 and Class 2 are similar, which makes it hard to distinguish.

2. In Figure 8, the exchange X-axis and Y-axis could be more intuitive.

Comments on the Quality of English Language

English is good enough for readers.

Author Response

REVIEWER 3

Comment 1: In this manuscript entitled “Interrogating Genomes and Geography to Unravel Multiyear Vesicular Stomatitis Epizootics”, J M Humphreys et al presented a comprehensive study to elucidate the spatial epidemiological patterns of the Vesicular Stomatitis New Jersey virus during the 2014-15 epizootic cycle in the United States. They found that a geographically defined Virus Pool at the US-Mexico interface was the source of VSNJV invasions and overwintering sites. Generally, the data sources are credible, the data analysis in this study is coherent, the inferences are proper and the results are meaningful and explicable.

Response 1: We thank the reviewer for the time spent evaluating the manuscript and the comments and suggestions provided.

Comment 2: The main deficiency is that the viral isolates from Mexico contain only 1 sample (n=1). Due to the high mutation in single-strand RNA viruses, too few samples may lack typicality.

Response 2: We fully concur with the reviewer that additional isolates from Northern Mexico are much needed to critically evaluate and expand on the results presented in the manuscript.  This was the principal reason for indicating that expanded field sampling and enhanced laboratory capacity are essential for further investigation (Lines 658-660).  Although the authors are actively identifying sources of additional samples, these data are unfortunately not yet available.  

Comment 3:

In Figure 7b, the VS case numbers sharply decreased beyond approximately 20 km distance from the river. Except for the vector insect reproduction and host animal behavior, does the amount of host animals also affect the VS case numbers? After all, most pastures are distributed along the river for an abundant water supply.

Response 3: Yes, the number of host animals does affect the number of VS cases.  Assuming all else equal, locations with more viable hosts are expected to be subject to more VS cases than locations with fewer available hosts.  Beyond the standard steps applied to avoid multicollinearity between host abundance and other variables,  the statistical model in the manuscript addresses this potential in several additional ways.  First, the model’s incorporation of preferential sampling (Lines 203-204) attempts to account for biased or uneven VS detection.  In this context, bias refers to areas with under- or over-representation in documented cases. Second, the model includes an additional spatial variable specifically to account for disease aggregations or clusters (Lines 221-223) that might be associated with abundant hosts.  Third, the occurrence probability and number of VS cases at a given location are proportional to, or “offset” by, the number of potential hosts (see “Exposure,” Lines 230-234).  As with all statistical analyses that do not incorporate formal causal analysis (Lines 558-561), variable interaction and confounding are a possibility.

Comment 4: In Figure 1, the lines indicating River Class 1 and Class 2 are similar, which makes it hard to distinguish.

Response 4: We agree that because Class 1 & 2 are only distinguished by line size/width, they may be challenging to distinguish in PDF/printed versions of the paper; however, this should be made easier through examination of the online version, which will allow for “zooming” in at higher resolution. Most importantly, the model treats both Class 1 & 2 locations equivalently. Therefore, there is no distinguishing done by the model or as part of interpreting the results.    

Comment 5: In Figure 8, the exchange X-axis and Y-axis could be more intuitive.

Response 5: We thank the reviewer for the suggestion; however, switching the axes would require the variable labels to be displayed vertically or at a steep angle across the X-axis, making reading them more difficult.  This may not be problematic if the variable names were shorter; however, in this case, some include multiple words.  We believe that keeping readable text along a flat, left-to-right arrangement is more appropriate for the figure.